# Analysis of Atmospheric and Ionospheric Variations Due to Impacts of Super Typhoon Mangkhut (1822) in the Northwest Pacific Ocean

**Mohamed Freeshah [1,2,3], Xiaohong Zhang [2,4,5,*], Erman Şentürk [6], Muhammad Arqim Adil [7], B. G. Mousa [1,8], Aqil Tariq [1], Xiaodong Ren [4,5] and Mervat Refaat [3]**

[1] State Key Laboratory of Information Engineering in Surveying, Mapping and Remote Sensing, Wuhan University, 129 Luoyu Road, Wuhan 430079, China; mohamedfreeshah@whu.edu.cn (M.F.); dr.ba haa1@azhar.edu.eg (B.G.M.); aqiltariq@whu.edu.cn (A.T.)

[2] Collaborative Innovation Center for Geospatial Technology, 129 Luoyu Road, Wuhan 430079, China; xhzhang@sgg.whu.edu.cn

[3] Department of Surveying Engineering, Faculty of Engineering at Shoubra, Benha University, 108 Shoubra St., Cairo 11629, Egypt; mervat.ameen@feng.bu.edu.eg

[4] School of Geodesy and Geomatics, Wuhan University, 129 Luoyu Road, Wuhan 430079, China; xdren@whu.edu.cn

[5] Key Laboratory of Geospace Environment and Geodesy, Ministry of Education, 129 Luoyu Road, Wuhan 430079, China

[6] Department of Surveying Engineering, Kocaeli University, Kocaeli 41001, Turkey; erman.senturk@kocaeli.edu.tr

[7] Department of GNSS, Institute of Space Technology, Islamabad 44000, Pakistan; arqim07@mail.ist.edu.pk

[8] Faculty of Engineering, Al-Azhar University, Cairo 11884, Egypt

* Correspondence: xhzhang@sgg.whu.edu.cn

**Abstract:** The Northwest Pacific Ocean (NWP) is one of the most vulnerable regions that has been hit by typhoons. In September 2018, Mangkhut was the 22nd Tropical Cyclone (TC) over the NWP regions (so, the event was numbered as 1822). In this paper, we investigated the highest amplitude ionospheric variations, along with the atmospheric anomalies, such as the sea-level pressure, Mangkhut's cloud system, and the meridional and zonal wind during the typhoon. Regional Ionosphere Maps (RIMs) were created through the Hong Kong Continuously Operating Reference Stations (HKCORS) and International GNSS Service (IGS) data around the area of Mangkhut typhoon. RIMs were utilized to analyze the ionospheric Total Electron Content (TEC) response over the maximum wind speed points (maximum spots) under the meticulous observations of the solar-terrestrial environment and geomagnetic storm indices. Ionospheric vertical TEC (VTEC) time sequences over the maximum spots are detected by three methods: interquartile range method (IQR), enhanced average difference (EAD), and range of ten days (RTD) during the super typhoon Mangkhut. The research findings indicated significant ionospheric variations over the maximum spots during this powerful tropical cyclone within a few hours before the extreme wind speed. Moreover, the ionosphere showed a positive response where the maximum VTEC amplitude variations coincided with the cyclone rainbands or typhoon edges rather than the center of the storm. The sea-level pressure tends to decrease around the typhoon periphery, and the highest ionospheric VTEC amplitude was observed when the low-pressure cell covers the largest area. The possible mechanism of the ionospheric response is based on strong convective cells that create the gravity waves over tropical cyclones. Moreover, the critical change state in the meridional wind happened on the same day of maximum ionospheric variations on the 256th day of the year (DOY 256). This comprehensive analysis suggests that the meridional winds and their resulting waves may contribute in one way or another to upper atmosphere-ionosphere coupling.

**Keywords:** atmospheric observations; tropical cyclones; ionospheric disturbances; typhoon Mangkhut; regional ionosphere maps (RIMs)

## 1. Introduction

Principally, geomagnetic storms and solar radiations play a crucial role in the dynamic regime of the ionosphere [1–4]. However, according to the evolution of the atmosphere-ionosphere coupling theory, the acoustic gravity waves (AGWs) could largely be associated with some powerful meteorological disturbances that further leads to some significant ionospheric perturbations [5–7]. Much research has been conducted to distinguish that AGWs could be associated with some powerful meteorological disturbances [8–10]. In 1960, the theoretical study of atmospheric AGWs have indicated that the ionosphere could respond to severe weather activities for instance lightning, cyclones, tornadoes, and hurricanes [11]. In addition, a recent paper claimed that large convective cells (typhoon is a perfect example of such a large-scale cell), which are the main generator of electricity in global electric circuit (GEC), lead to the local changes of the ionospheric potential [12]. Both convective activity and the area covered by electrified clouds are dominant phenomena for ionospheric potential parameterization.

As it is known, the tropical cyclones (TCs) have powerful vortical structures in such a way that their origin occupies some near-equator regions from 5º and 20º in latitudes, in each hemisphere [13,14]. In Southeastern Asia, TCs are referred to as typhoons whenever become at the hurricane state [15]. According to the international classification, the stages of a TC's development could be categorized based on sustained maximum wind speed in a cyclone: (1) the maximum sustained wind speed under 61 km/h is called tropical depression, (2) about 65–115 km/h is called a tropical storm, and, (3) if it gets over 119 km/s, it will be called as a typhoon in Southeastern Asia and referred to as a hurricane in the Atlantic Ocean and the northeastern Pacific Ocean [15–17].

For the first time, Bauer has reported the findings of the ionospheric response to typhoon passage where the F2 layer's critical frequency, the so-called foF2, was increased when the typhoon getting close to the High-Frequency (HF) Radio station [18]. However, the foF2 data has low spatial resolution with sparse data on the globe [19,20]. An internal atmospheric wave (IAWs) could be generated having a period of 1–150 min as a result of robust tropospheric disturbances from TCs. Under favorable conditions, the IAWs can pierce into the ionosphere causing ionospheric disturbances [21–23]. Besides, the electric field and ion oscillation in a studied region would enhance according to the strength of the cyclone, and this depends on charged aerosols and droplets [24,25]. On the same side, Sorokin et al. (2005) and Pulinets et al. (2000) reinforced that an ionospheric plasma irregularity may be generated by the electric fields over the regions of powerful synoptic perturbations [26,27].

The lower side of the ionosphere is more sensitive to tropospheric activities [22,23], whereas recording the upper ionospheric (F region) perturbations, caused by the typhoon, faces several difficulties due to the intensity weakness of the ionospheric response and large ionospheric vulnerability to geomagnetic effects at those altitudes. However, ionospheric perturbations, associated with the typhoon effect on the upper side of the ionosphere, were mainly revealed by observing the Faraday rotation of the polarization plane or the Doppler frequency shift [28–31]. According to previous studies, the Doppler sounding method has false oscillations, where it causes ionospheric disturbances by itself, and it could be automatically transformed to time series data [32]. So, HF Doppler sounders is not a very good technique to identify the ionospheric variations induced by cyclones [30,33].

Recently, the Global Navigation Satellite System (GNSS) provides global coverage Total Electron Content (TEC) data with high temporal and spatial resolutions [34]. By using GNSS observations, the TEC values could be calculated over a ground GNSS station then the ionospheric response to typhoon could be studied [35]. The highest amplitudes of slant TEC (STEC) or vertical TEC (VTEC) variations have been recorded during the cyclone's maximum wind speed (maximum spot) [14]. Many previous studies have been concentrated more on ionospheric variations associated with TCs on the day of typhoon landfall and neglected the maximum spot [6,36–38]. On the other side, some researchers

focus on the ionospheric response to TCs at evening/night local hours [13,14,38]. Li et al. (2017) used the VTEC maps from the Center for Orbit Determination in Europe (CODE) to detect the ionospheric disturbances during the passage of powerful hurricanes and typhoons in East Asia and North America. However, Global ionosphere maps (GIMs) have a low spatial resolution of 2.5° in latitude and 5° in longitude [39]; moreover, the VTEC data depends mainly on the International GNSS Service (IGS) stations and lacks the dense Continuously Operating Reference Stations (CORS) stations, which makes it have low spatial resolution, especially for regional case studies. Li et al. (2017) also neglected the atmospheric parameters, except for satellite cloud photography.

Different mechanisms have been proposed to understand the ionospheric variations caused by the TCs. Since the typhoons carry rain clouds and thunderstorms, Wilson (1920) proposed a thunderstorm generator hypothesis by explaining that the thunderstorms deposit negative charges to the Earth and positive charges to the upper atmosphere [40]. Similarly, Rycroft et al. (2012) proposed a GEC model that explains the generation of the vertical electric field in the atmosphere and the creation of the potential difference between Earth's surface and the lower ionosphere due to the thunderstorm and electrified convective clouds activities [41]. During thunderstorm activity, the ionosphere acquires a positive potential between 200–500 kV, compared to the Earth's surface. They further explained that this potential difference drives down the vertical conduction current towards the ground from the ionosphere in all Earth's fair-weather regions. The fair-weather current depends upon the potential difference and the columnar resistance between the ground and the ionosphere. Furthermore, the horizontal currents flow independently in the highly conducting ionosphere and the surface of the Earth. The upward flow of the current towards the ionosphere from the thunderstorm cloud top and from the ground to the thunderstorm closes the circuit. On the other hand, the propagation and evolution of the atmospheric AGWs depend upon the propagation medium and its properties [11]. The altitudinal exponential decrease in the atmospheric pressure generates the upward propagation of the AGWs. The amplitude of the wave is proportional to the atmospheric density, however, the amplitude of the AGWs enhances to maintain the constant energy flux according to the energy conservation law [11]. At an altitude of ~150 km, the resulting amplification factor reaches $10^4$, whereas, between 350–400 km, this factor reaches $10^5$-$10^6$. At these altitudes, the AGWs produces significant perturbation in the ionosphere plasma through photochemical and dynamical processes.

In this contribution, we produced the regional ionospheric maps with a spatial grid 0.2° in latitude and 0.4° in longitude with 2 h temporal resolution to coincide with CODE by using dense CORS stations of Hong Kong (HKCORS) in addition to temporal VTEC from IGS stations around the area of interest. Finally, the atmospheric parameters, such as sea-level pressure and the meridional and zonal winds, were retrieved from National Oceanic and Atmospheric Administration-Physical Sciences Laboratory (NOAA-PSL) to analyze their possible impacts on ionospheric response through the anomaly maps of the sea level pressure, zonal wind, and meridional wind.

In this paper, we extensively used acronyms throughout the whole text. To make a proper way to find the corresponding meaning, we listed all used acronyms in alphabetical order in Table 1.

**Table 1.** Lists of acronyms and abbreviations.

| Acronyms | Corresponding Meaning |
|---|---|
| AGWs | Acoustic Gravity Waves |
| $B_L$ | Lower Bound |
| $B_U$ | Upper Bound |
| CCL | Carrier-to-Code Leveling |
| CODE | Center for Orbit Determination in Europe |
| CORS | Continuously Operating Reference Stations |
| DOY | Day Of the Year |

| Dst | Disturbance storm-time |
|---|---|
| EAD | Enhanced Average Difference |
| F10.7 | solar radio flux at 10.7 cm |
| GEC | Global Electric Circuit |
| GIMs | Global Ionosphere Maps |
| GNSS | Global Navigation Satellite System |
| HF | High-Frequency |
| HK | Hong Kong |
| HKT | Hong Kong Time |
| IAWs | Internal Atmospheric Waves |
| IGS | International GNSS Service |
| IPP | Ionospheric Pierce Point |
| IQR | InterQuartile Range |
| Kp | the geomagnetic Kp |
| LOS | Line Of Sight |
| MNG | mean of the nearest two grid points |
| NOAA-PSL | National Oceanic and Atmospheric Administration-Physical Sciences Laboratory |
| nT | nano Tesla |
| NWP | Northwest Pacific Ocean |
| R1 | lowest range limit |
| R2 | highest range limit |
| RIMs | Regional Ionosphere Maps |
| RTD | Range of ten Days |
| sfu | Solar Flux Unit |
| SLM | single-layer mapping |
| STEC | Slant Total Electron Content |
| TC | Tropical Cyclone |
| TEC | Total Electron Content |
| TECU | Total Electron Content Unit |
| VTEC | vertical Total Electron Content |

## 2. Data Sources and Analysis Methods

*2.1. Space Environment and Super Typhoon Mangkhut Data*

According to the recorded data for the maximum sustained wind near the center of the low-pressure area of (Mangkhut), there are two points with maximum wind speed at all as follows in Table 2.

**Table 2.** The geographical location of the maximum sustained wind speed point in km/h, besides Hong Kong Time (HKT), date of the year (DOY), and the typhoon classification.

| Time (HKT) | Days of September | DOY | Geographical Location (degree) | | Classification | Maximum sustained wind speed (km/h) |
|---|---|---|---|---|---|---|
| 2:00 | 11 | 254 | 14.0 N | 142.6 E | Severe Typhoon | 165 |
| 8:00 | 11 | 254 | 14.0 N | 141.2 E | Super Typhoon | 185 |
| 14:00 | 11 | 254 | 13.9 N | 139.8 E | Super Typhoon | 205 |
| 20:00 | 11 | 254 | 13.7 N | 138.6 E | Super Typhoon | 220 |
| 8:00 | 12 | 255 | 13.9 N | 136.2 E | Super Typhoon | 240 |
| 2:00 | 13 | 256 | 14.4 N | 132.5 E | Super Typhoon | 240 |
| 2:00 | 14 | 257 | 15.2 N | 127.9 E | Super Typhoon | 240 |
| 20:00 | 14 | 257 | 17.4 N | 124.1 E | Super Typhoon | 240 |
| 23:00 | 14 | 257 | 17.7 N | 123.2 E | Super Typhoon | 250 |
| 2:00 | 15 | 258 | 18.0 N | 122.3 E | Super Typhoon | 250 |
| 5:00 | 15 | 258 | 18.0 N | 121.3 E | Super Typhoon | 230 |
| 11:00 | 15 | 258 | 18.3 N | 120.1 E | Super Typhoon | 195 |

As seen in Table 2 and Figure 1, the maximum wind speed is 250 km/h over the location of (17.7° N, 123.2° E) at 23:00 Hong Kong Time (HKT) on 14 September 2018 and over the location of (18.0° N, 122.3° E) at 02:00 HKT on 15 September 2018. Previously, Li et al. (2017) have interpolated the VTEC values of the four grid points nearest to the maximum spot and apply the method developed by Schaer, (1999) to get the VTEC values [42]. In this study, we have computed the VTEC time series over the maximum spots as the mean of the nearest two points, where each point is located on a grid line between two grid points, and the grid size of the created RIMs is narrower than GIMs which provided high spatial resolution VTEC data.

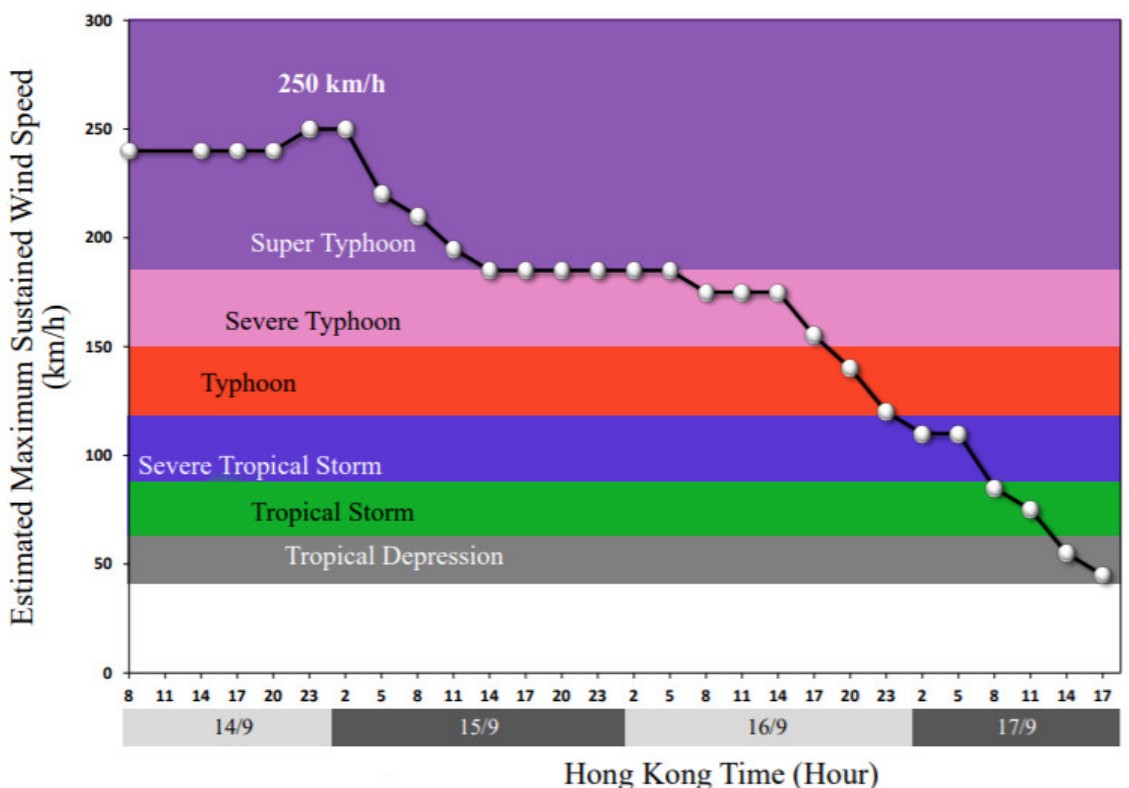

**Figure 1.** Time series of maximum sustained wind speed of Mangkhut from September 14th–17th 2018 with colors representing the typhoon classification. (https://www.hko.gov.hk/en/informtc/mangkhut18/maxWind.htm).

*2.2. Atmospheric Parameters*

The atmospheric parameters were retrieved from NOAA-PSL via http://psl.noaa.gov/data/-composites/day. These data set have a temporal coverage of daily mean values and spatial coverage of 2.5 x 2.5-degree global grids (144x73) between 90° N-90° S latitudes and 0° E-357.5° E longitudes [43]. The anomaly maps of each observed day represent differences from the mean value of the previous 5 days. In this study, the anomaly maps of the sea-level pressure, zonal and meridional winds for a region between 0-30° N and 100-145° E from September 10–17, 2018 (from DOY 253 to DOY 261), were used. On the other side, the infrared satellite snapshots of typhoon Mangkhut's cloud system were selected for certain moments from the Cooperative Institute for Meteorological Satellite Studies / University of Wisconsin-Madison via http://tropic.ssec.wisc.edu/archive/.

*2.3. Calculation Process of STEC, VTEC, and RIMs*

Many researchers have utilized the data of the GIM assimilative model. Globally, the data of more than 400 stations are continuously collected to produce GIM pattern, subsequently, generates ionospheric VTEC maps through interpolation and smoothing techniques which extend from ±87.5 latitudes and ±180 longitudes with spatial resolutions of 2.5 and 5 degrees, respectively [44]. As we can see, produced VTEC maps through GIMs not only have low spatial resolutions but also depend on limited stations, especially, in the area of interest around Super Typhoon Mangkhut. Previous studies indicated that the intensity of tropical typhoons may affect the occurrence rate of ionospheric disturbances [14]. For that, we collected the GNSS observations from IGS and the HKCORS in the area around the typhoon. The HKCORS contains 18 CORS evenly set up in Hong Kong for more details we can refer to (https://www.geodetic.gov.hk/en/satref/satref.htm). Therefore, regional ionospheric maps (RIMs) with a high spatial resolution of 0.4 x 0.2 degrees in longitude and latitude, respectively, have been created through the collected observation with a temporal resolution of 2 h to coincide with GIMs. The calculated RIMs cover 110°-130° E longitudes and 10°-25° N latitudes. This provides us a very good opportunity to investigate the ionospheric variations associated with the typhoon Mangkhut by using high spatial resolution maps.

In this study, the produced RIMs are used to investigate the possible ionospheric disturbances during the passage of the powerful typhoon Mangkhut over the points of maximum wind speed [45]. The ionospheric response to the typhoon was analyzed after computing the ionospheric information. We derived the Slant TEC (STEC) along the line of sight (LOS) between receivers to GNSS satellites. We can express the ionospheric delay in length unit and TEC as a function in ʃ frequency (f1 = 1575.42 MHz, f2 = 1227.60 MHz) [46,47] through the equation:

$$STEC = \frac{I \times f^2}{40.31 \times 10^{16}},\tag{1}$$

where $I$ is the ionospheric delay along a LOS. The GNSS measurements encompass two L-band frequencies for the carrier phase and pseudo-range. The carrier-to-code leveling (CCL) technique is carried out for each continuous arc where the CCL degrades the pseudo-range noise, and it eliminates potential ambiguity influence, as well as it retains the high precision in the carrier-phase [48]. The CCL method utilizes geometry-free linear combinations of code and phase observations to derive the information of the ionosphere [44,45]. Recently, Zhang et al. (2014) showed that the geometry-free combination is more sensitive to ionospheric activity, particularly in the existence of ionospheric scintillation. The final CCL algorithm form for the processed ionospheric observable can be expressed through the equation:

$$\tilde{L}_{I,arc} = L_{I,arc} - \left(L_{I,arc} - P_I\right)_{arc} = STEC + B_r + B^S + \left(\varepsilon_p\right)_{arc} + \varepsilon_L,\tag{2}$$

where $\tilde{L}_{I,arc}$ is the carrier-phase ionospheric observable leveled to the code-delay one, and $L_{I,arc}$ is the carrier-phase ionospheric observable; $B_r$ and $B^S$ are the receiver and satellite hardware delays of the pseudo-range code, respectively. $\varepsilon_L$ and $\varepsilon_P$ are the noise and multipath for the carrier-phase ionospheric observations and for the code-delay measurements, respectively.

Frequently, TEC is computed by single-layer mapping (SLM) because the electron density has a complex spatial distribution [49]. The SLM is used to convert STEC into VTEC using the mapping function. However, the low elevation angle part is a common factor that caused errors in different mapping function methods. A lower elevation angle will cause a larger mapping function error, and higher elevation angle will cause data loss. In this paper, the elevation cut-off angle was set to be 15 degrees, which can reduce the multipath error, on the other hand, can control the mapping error. According to the modeling-hypothesis, an SLM where the 2D modeling process assumes that the total free electrons of the whole ionosphere is concentrated in a thin layer at the elevation with the max

electron density [50,51]. The STEC value along the LOS at the Ionospheric Pierce Point (IPP) can be converted into the corresponding VTEC by using a common ionospheric mapping function.

$$VTEC = \cos\left[\arcsin\left(\frac{R}{R+H}sinz\right)\right]STEC, \tag{3}$$

where R represents the average radius of the earth, H denotes the altitude of the ionospheric shell layer, and z stands for the satellite zenith angle at the location of the receiver.

We have created a high resolution regional ionospheric maps (RIMs) around the area of interest, where it can be expressed by the spherical harmonics' expansion as follows:

$$VTEC(\beta, s) = \sum_{n=0}^{nmax}\sum_{m=0}^{n}\tilde{P}_{nm}(sin\beta)(A_{nm}\cos ms + B_{nm}\sin ms), \tag{4}$$

where s denotes the solar-fixed longitude, and β represents the geomagnetic latitude of the IPPs, s = λ - λ₀, where $\lambda_0$ and λ denote the longitude of the sun and the IPP, respectively. $\tilde{P}_{nm} = \Lambda$ (n, m) $P_{nm}$ represents a normalized associated Legendre polynomial of degree n and order m; $A_{nm}$ and $B_{nm}$ are spherical harmonic coefficients to be estimated. $P_{nm}$ is the non-normalized associated Legendre polynomial, and Λ (n, m) is the normalization constant can be defined as:

$$\Lambda(n, m) = \sqrt{(n-m)!(2n+1)(2-\delta_{0m})(n+m)!}, \tag{5}$$

where $\delta_{0m}$ is the Kronecker delta symbol [34,50].

In this study, we are more concerned about the entire ionospheric disturbance during the typhoon, rather than a certain layer [49]. The ionospheric single-shell model with 4th order and 4th degree for the "Spherical Harmonic" expansion was taken to create the RIMs for 17 days with 0.4 degrees in longitudes and 0.2 degrees in latitudes. To reduce the multipath error, the elevation cut-off angle was set to be 15 degrees. In addition, the carrier-phase ambiguity will stay fixed as long as the satellite antenna is not reoriented (e.g., during the wind-up); therefore, the term of carrier phase wind-up has been adjusted by orbital information and coordinates of each station.

*2.4. Methods For Detecting VTEC Disturbances.*

In this paper, we proposed a method to compute the VTEC time series values over maximum wind speed points, so-called the "maximum spots", as the mean of the nearest two grid points (MNG), where the grid size in the created RIMs is relatively small, and each point of maximum spots located on a grid line.

The VTEC time series over the two maximum spots for 17 days were analyzed by the interquartile range (IQR) approach. To discover the VTEC variations, the VTEC values of 10 days before September 11, 2018, were used as window length to calculate the upper quartile, lower quartile, and median, denoted as $Q_U$, $Q_L$, and M, respectively.

During the process of data analysis, the values of VTECs over the two maximum spots at the same time were extracted from the RIMs by the proposed MNG method and ranked from the lowest to the highest in ascending order, i.e., $Y_1$, $Y_2$, . . .$Y_{10}$. Then, we can calculate QU, QL, M, and IQR as follows:

$$Q_L = \frac{(Y_2 + Y_3)}{2}, \tag{6}$$

$$Q_U = \frac{(Y_8 + Y_9)}{2}, \tag{7}$$

$$M = \frac{(Y_5 + Y_6)}{2}, \tag{8}$$

$$IQR = Q_U - Q_L, \tag{9}$$

By applying the double of IQR as a tolerance (~2.7 standard deviations) [52], we can calculate the upper and lower bounds $B_U$ and $B_L$, respectively, as follows:

$$B_U = M + 2 * IQR, \qquad (10)$$

$$B_L = M - 2 * IQR, \qquad (11)$$

Either of the VTECs values falling outside of the corresponding upper or lower bounds are regarded as abnormal signals. These VTEC anomalies can last for 6 h at least [33]. According to Hong Kong Observatory, the first and second maximum spots have coordinates of 17.7° N - 123.2° E and 18.0° N - 122.3° E, respectively. The VTEC time series over the two maximum spots were computed. Then, the VTEC time series were analyzed for the next seven days (from DOY 254 to DOY 260) during Super Typhoon Mangkhut over the two maximum spots.

For a better understanding of how is the data set varied, and try to find key measures of the detected ionospheric disturbances, a second method was applied, the so-called range of ten days (RTD), the VTEC values of the 10 days' data set were ordered to calculate the range, and then the highest and lowest range limit's values in the set were compared with the detected ionospheric disturbances' DOY.

In order to weaken or eliminate the effects of periodic variations, we applied a third de-trending method. In 2020, Freeshah et al. proposed a de-trending method called one-week average (AD) by taking the average of STEC for one week and subtract from STEC values for the day of the thunderstorm over a mid-latitude region [53]. We enhanced the AD approach (EAD) by increasing the number of days for the estimated average value to be ten days; then, we applied the EAD for VTEC values over the northwest Pacific Ocean (a low-latitude region), to detect the ionospheric disturbances over the maximum spots by using the VTEC time series. In order to avoid the possible geomagnetic effect that—as a result of the increase of geomagnetic Kp index in these two days—could perturb VTEC on DOY 254 and DOY 255, we excluded these two DOYs from the average value. In addition, in order to make the average value more reliable, we selected a period of ten days, referred to as the time that preceded the typhoon transformation into the super typhoon occurred on September 11, 2018. Then, we calculated the average value during these ten days. In fact, it is reasonable to assume that these ten days can represent the normal state of the ionosphere in this period.

The EAD method is employed to detect ionospheric disturbances based on an average of ten days before the tropical cyclone converted to a super typhoon. The subtraction of the average simultaneous observations from the next 7 days can lead to an observable residual. To check the highest VTEC variations among 7 days started from the first day of the super typhoon, the residual could be calculated from the difference between the observed value and the estimated (average) value of the ten days, where the highest residual values could reflect the highest deviation of VTEC. The mathematical process for the EAD method of de-trending can be described as follows:

$$VTEC_{av} = (VTEC_{Doy244} + VTEC_{Doy245} + \cdots + VTEC_{Doy253})/10, \qquad (12)$$

$$\delta VTEC_{res} = VTEC_{Doy(i)} - VTEC_{av}, \qquad (13)$$

where $VTEC_{av}$ denotes the average VTEC for 10 days, $\delta VTEC_{res}$ represents the residuals in VTEC values, and $VTEC_{Doy(i)}$ indicates a specific day from DOY 254 to DOY 260. The residuals for 7 days over the maximum spots during typhoon were then compared.

## 3. Results and Discussion

### 3.1. Geomagnetic Field and Solar-Terrestrial Environment

Geomagnetic activities and solar radiation are the primary influencers of the ionospheric variations [10]. To distinguish whether the space weather impacts on VTEC deviations during the typhoon, the solar radio flux at 10.7 cm (F10.7), the Disturbance storm-time (Dst), and the geomagnetic Kp indices have been investigated (Figure 2). The geomagnetic field intensity of the earth can be categorized as three levels: low (Dst > -50 nT),

moderate (-100 nT < Dst ≤ -50 nT), and high (≤-100 nT), where nT is expressed in nano-Tesla [54]. In contrast, solar intensities could be categorized into four intensity levels as follow: low (F10.7 < 100 sfu), moderate (100 sfu ≤ F10.7 < 150 sfu), high (150 sfu ≤ F10.7 < 200 sfu), and extreme (F10.7 ≥ 200 sfu), where sfu is solar flux unit: 1 sfu = $10^4$ Jy = $10^{-22}$ W·m$^{-2}$·Hz$^{-1}$ = $10^{-19}$ erg·s$^{-1}$·cm$^{-2}$·Hz$^{-1}$ [55].

As seen in Figure 2, during the whole period, any evidence of a strong solar activity or geomagnetic activity is not observed. The lower panel of Figure 2 shows the Dst index ranging between +20 to -20 nT during the typhoon Mangkhut, except for DOY 254 and DOY 255 (the two excluded days from the IQR window length and the EAD method), indicating that geomagnetic activity was relatively in steady-state for the period of data. As shown in the upper panel, the F10.7 index indicated that the solar activity can be considered as low-level during the typhoon Mangkhut where F10.7 values were less than 100 sfu. There is a low geomagnetic activity (Kp ≤ 4), except on the DOY 257 (Kp ≤ 4.3), DOY 253 (Kp ≤ 5), and the DOY 254 that has the highest value of Kp ≤ 6, indicating a moderate geomagnetic activity, DOY 254 was excluded from the IQR window length and the EAD method to skip the possible variation based on moderate geomagnetic activity. Figure 2 indicates that the solar radiation and geomagnetic activity component were relatively stable ten days before the typhoon. It is warranted that the ionospheric conditions before the typhoon, by using the IQR method and EAD de-trending method, were not affected by solar and geomagnetic activities, and the results of these methods will be shown later in Section 3.2.

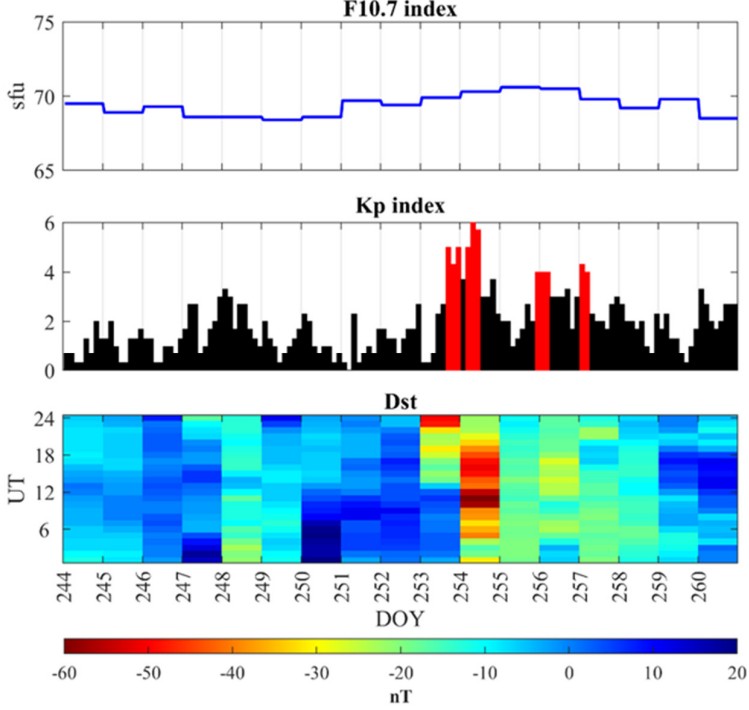

**Figure 2.** The F10.7, Kp, and Disturbance storm-time (Dst) indices in the period from DOY 244 to DOY 260.

*3.2. Ionospheric Variations During Super Typhoon Mangkhut*

Figure 3 shows the VTEC time series for 24 h/week over the 1st and 2nd max wind speed points of the typhoon. The VTEC is expressed by total electron content unit (TECU), where: 1 TECU = $10^{16}$ electrons/m$^2$. As shown in Figure 3a, the range of 34.55–1.2 TECU (highest VTEC and lowest VTEC) was maintained for most of the time during the typhoon except for DOY 256, where the range increased significantly to become 41.9–4.2 TECU. For

DOY 256, VTEC gradually increased to the maximum value of about 08:00 UT (16:00 HKT), and then VTEC gradually decreased. The VTEC anomalous variations can be seen clearly after 06:00 UT (14:00 HKT). As shown in Figure 3b, the range of 34.6–1.25 TECU was maintained for most of the time during the typhoon except for DOY 256, where the range increased significantly to become 42.05–4.35 TECU. For DOY 256, VTEC gradually increased to the maximum value of about 08:00 UT (16:00 HKT), and then VTEC gradually decreased. The VTEC anomalous variations can be seen clearly after 06:00 UT (14:00 HKT). Fig. 3 indicates that the deviation of VTEC values of DOY 256 over the two max wind speed points could arrive up to more than 7 TECU as a large variation occurred during 7 days.

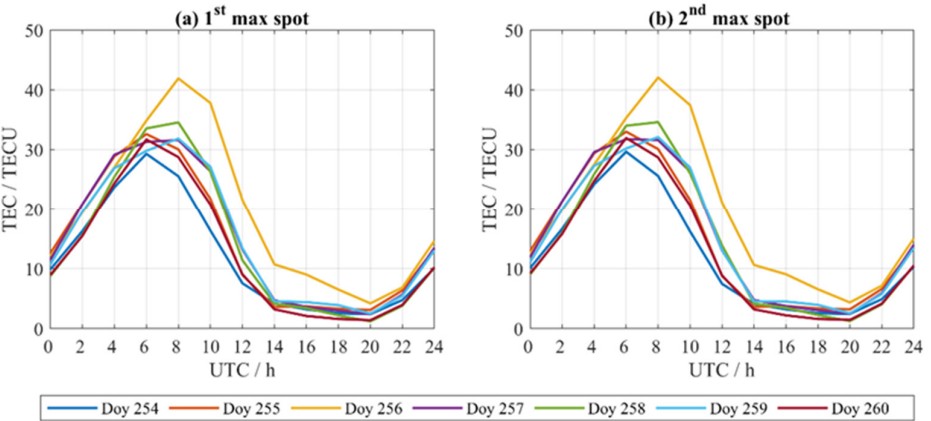

**Figure 3.** Vertical Total Electron Content (VTEC) time series over (**a**) 1st and (**b**) 2nd maximum spots of the Super Typhoon Mangkhut, respectively, from September 11–17, 2018 (DOY 254–DOY 260).

According to the Geostationary Operational Environmental Satellite data (https://hesperia.gsfc.nasa.gov/goes/goes_event_listings/), there is a brief solar flare that happened during the daytime for three days of the analyzed data (11, 13, and 16 September 2020); see Table 3.

**Table 3.** The start, peak, and end time (UTC) of Solar Flares from the Geostationary Operational Environmental Satellite from September 11–17, 2018 (DOY 254–DOY 260).

| Date | DOY | Start Time | Peak Time | End Time | Class |
|---|---|---|---|---|---|
| Sept. 11 | DOY 254 | 7:56 | 7:59 | 8:01 | B |
| Sept. 12 | DOY 255 | NA | NA | NA | NA |
| Sept. 13 | DOY 256 | 17:56 | 17:58 | 18:02 | A |
| Sept. 14 | DOY 257 | NA | NA | NA | NA |
| Sept. 15 | DOY 258 | NA | NA | NA | NA |
| Sept. 16 | DOY 259 | 4:34 | 4:35 | 4:36 | A |
| Sept. 17 | DOY 260 | NA | NA | NA | NA |

To check if there is such an effect on VTEC variations that may be associated with solar flares, Figure 3 shows that the VTEC of 11 and 16 September is close to the other days without solar flares, and it has almost the same behavior where the solar flares happened for a very short time, 2–6 min, and only for one time a day. In this sense, solar flares could cause transient ionospheric disturbances by changing electron density, but, once the action of the solar flare is over, the disturbed plasma tends to the original state [56]. Similarly, the day of maximum variations on 13 September (DOY 256) also may not be perturbed by solar flares, not only for short time solar flare, but also the time of solar flare

that started at 17:56 and continued until 18:02 UTC, after more than 9 h of the maximum VTEC variations at 08:00 UTC. It is worth indicating that the solar flares class are very weak (A and/or B classes), and the influences of these flares are not important. So, the VTEC disturbances on DOY 256 are induced by the typhoon rather than by other random events.

Figure 4 shows the VTEC time series of DOY 256 versus the lower and upper bounds in blue and orange colors, respectively. Figure 3a,b showed a significant increase in VTECs over the two maximum wind speed points last about 6–14 UT. The lower and upper bounds used to distinguish the ionospheric anomalies response to typhoon depend on the IQR through the ten previous days before the cyclone converted into a super typhoon. It is worth indicating that the ionospheric disturbances have a positive value and agree with the general behavior, shown in Figure 3.

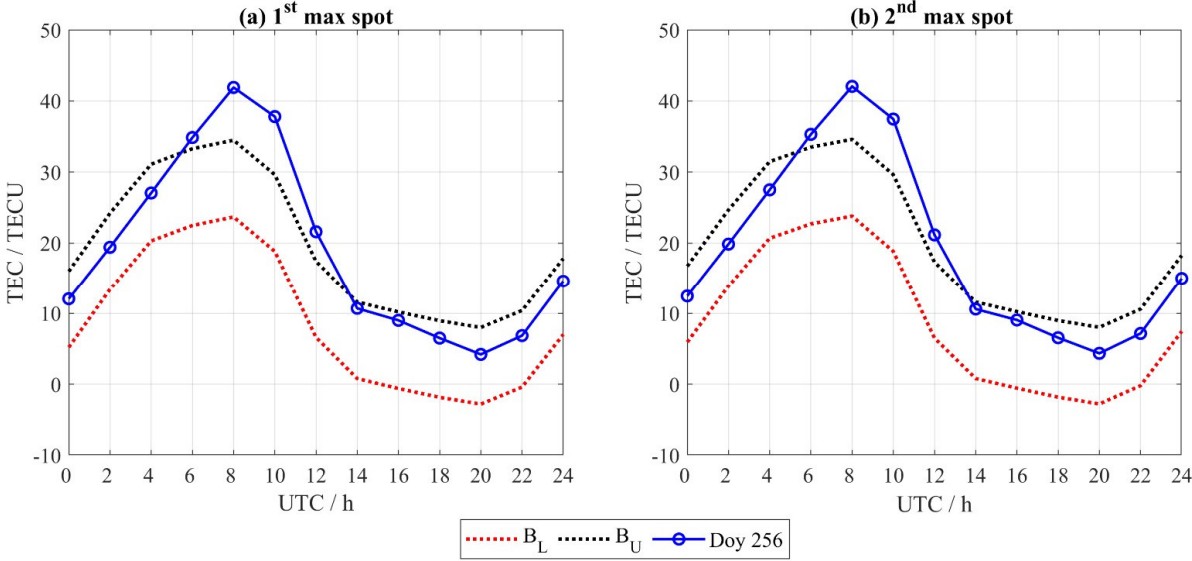

**Figure 4. V**TECs lower (B$_L$) and upper (B$_U$) bounds of the interquartile range method (IQR) method versus the VTEC time series of DOY 256. The data represented over (**a**) 1st and (**b**) 2nd maximum spots of the Super Typhoon Mangkhut from DOY 244 to DOY 253.

Figure 5 shows the results of by RTD method, where the VTEC time series of DOY 256 versus the lowest and highest range limits values overall in the ten days before the super typhoon. As it is known, the range of a data set is considered as a substantial measurement, where the figures at the bottom and top of it stand for the findings furthest removed from the majority. The lowest and highest range limits in ten days indicate the maximum and minimum of VTEC sequences during the normal behavior before the typhoon. As seen in Figure 5, the VTEC time series range of the ten days before the typhoon is clustered around the minimum and maximum. The VTEC time series in DOY 256 has exceeded the highest range limit during the specific period and arrives at the maximum value around 08:00 UT; then, VTEC data go to degradation, the disturbance almost ended around 14:00 UT, and the differences after 14:00 UT are not significant. So, it could be considered as VTEC ionospheric disturbance when its values override the highest range limit from 6–14 UT. During the ionospheric anomalous deviations from about 6–14 UT, the maximum difference between DOY 256 VTEC values and highest (R2) range limit were 8.25 TECU and 7.95 TECU over the (a) 1st and (b) 2nd maximum spots of the typhoon, respectively. Logically, the results over the two maximum spots are convergent where the two points of maximum sustained wind speed away with only ~100 km.

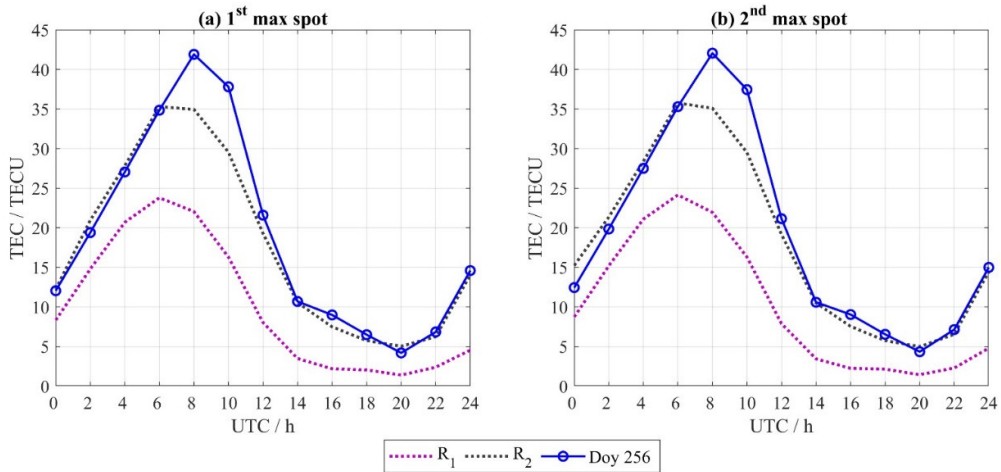

**Figure 5.** VTECs, lowest (R1) and highest (R2) range limits of range of ten days (RTD) method versus VTEC time series of DOY 256. The data represented over (**a**) 1st and (**b**) 2nd maximum spots of the typhoon from DOY 244 to DOY 253.

Figure 6 indicated the results of the EAD de-trending technique to detect the VTEC ionospheric variations during the typhoon over the maximum spots. The differences between VTEC values and the average of ten days before the typhoon are depicted below. As seen, most of the days from DOY 254 to DOY 260 show small residuals that means the values in the statistical data set are close to the average of the 10 days' VTEC time series. Except for DOY 254, there are VTEC variations from around 8–12 UT mainly caused by the geomagnetic activity where DOY 254 has the highest value of $Kp \leq 6$ (Fig. 2). However, these values are insignificant relatively compared to the VTEC variations on DOY 256. Fig. 6 shows the significant deviations from ~6-14 UT where the maximum residual value was at about 10:00 UT. The large residuals during DOY 256 indicated that the values in the VTEC time series are farther away from the average value of the specified period of 10 days. According to the previous study, the GNSS-VTEC GIMs' standard deviations are about 0.7–4.5 TECU [57]. As it is known that GIMs have a global scale and may not as accurate as of the derived RIMs [58]. However, we can consider RIMs have the same GIMs accuracy in the worst case. All days' residuals are within the width of the RIMs accuracy, except for DOY 256, which has significant deviations. In addition, the large residuals during the peak hours ~6–14 UT of DOY 256 reflect a large amount of variation in VTEC values and confirm the results of the previous methods which indicated that the ionospheric response to super typhoon on DOY 256 during the same period.

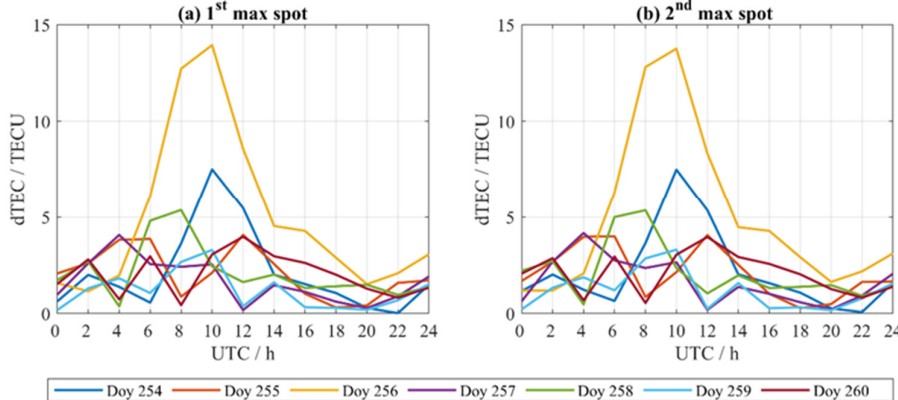

**Figure 6.** VTEC residuals for revealed DOY 256 among a week of the Super Typhoon Mangkhut from DOY 254 to DOY 260 over (**a**) 1st and (**b**) 2nd maximum spots.



### 3.3. Atmospheric Observations During Super Typhoon Mangkhut

To show the effects of the typhoon Mangkhut on the atmospheric climate and try to enhance the understanding of the troposphere-ionosphere coupling, the atmospheric parameters during the typhoon were retrieved from NOAA-PSL with 2.5x2.5 degrees spatial resolution. The anomaly maps of days represent differences from the average value of the previous 5 days. The maps were obtained from September 10–18, 2018 (DOY 253–DOY 261).

Figure 7 shows that the behavior of the typhoon in the zonal flow's direction tends to be weaker and has fast-moving which causes a relatively slight impact on the local weather. Meanwhile, Figure 8 shows the meridional flows tend to be stronger with slow-moving. This pattern is accountable for most instances of severe weather during the typhoon. This kind of extreme weather during flow regime is not only marked with strong storms but also the temperatures can reach their extremes, these weather disturbances can produce cold waves and heat waves depending on the poleward or equator-ward direction of the flow [59,60]. By returning to Figure 8a–c, we can see the meridional wind has slight variations from DOY 253–DOY 255. Then, a critical change has happened from DOY 255 (panel c) to DOY 256 (panel d), and this variation indicates a sudden increase in the meridional wind between the aforementioned DOYs. It is worth indicating that the critical change of meridional wind happened on the same day of maximum ionospheric variations DOY 256. This may be a possible reason that the meridional winds and their resulting waves may contribute to upper atmospheric coupling.

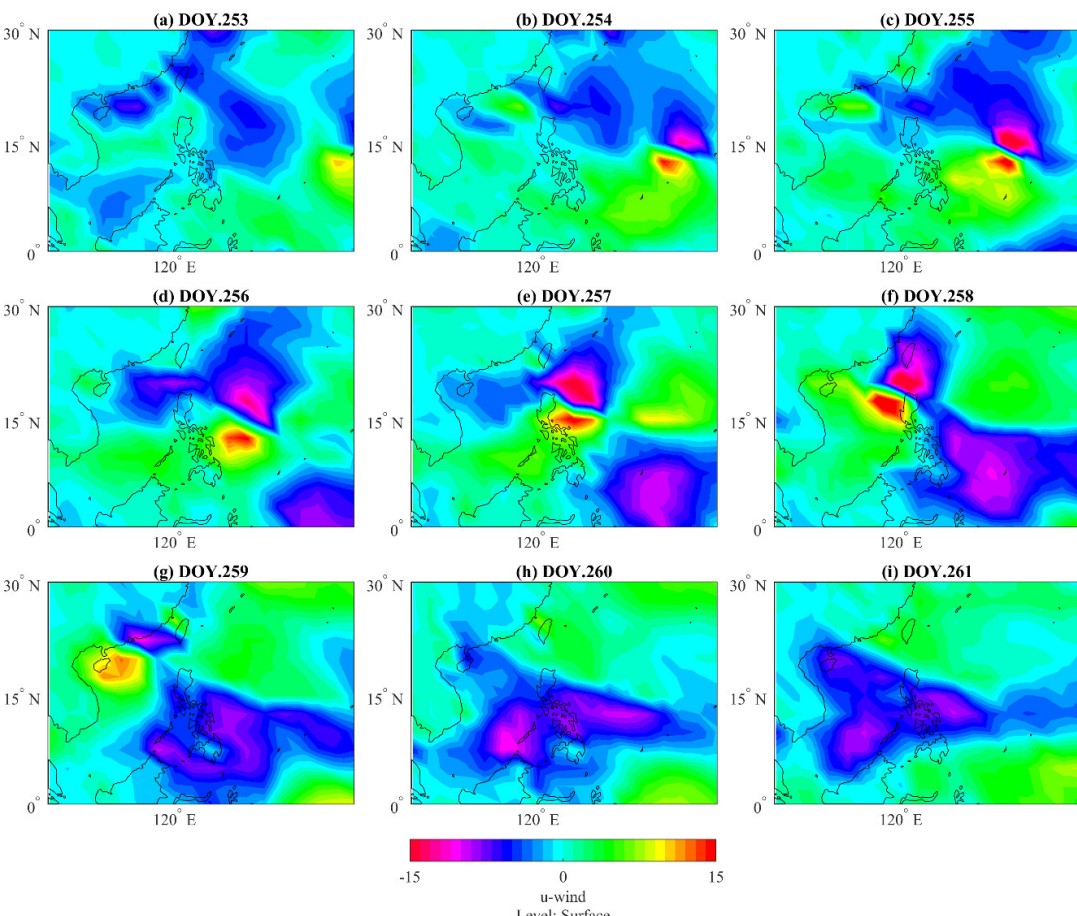

**Figure 7.** The anomaly maps of zonal wind from September 10–18, 2018; the panels (**a**)–(**i**) corresponds to DOY 253–DOY 261, respectively.

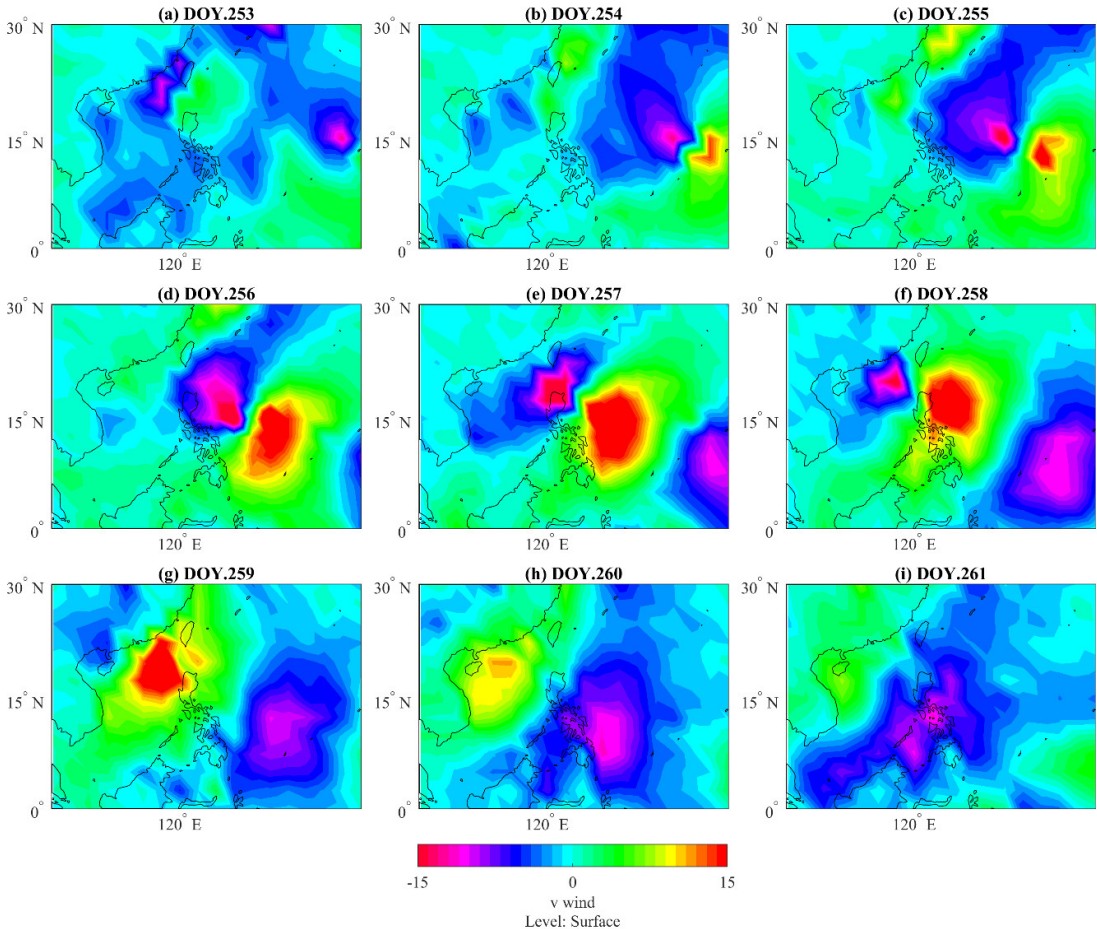

**Figure 8.** The anomaly maps of meridional wind on September 10–18, 2018; the panels (**a**)–(**i**) corresponds to DOY 253–DOY 261, respectively.

Figure 9 shows sea-level pressure from September 10–18, 2018 (DOY 253–DOY 261). The sea level pressure tends to decrease around the typhoon periphery on DOY 257 (panel e) and DOY 258 (panel f), and the low-pressure cell covers the largest area on DOY 256 (panel d). Therefore, the highest ionospheric VTEC disturbance amplitude can be observed when the low-pressure cell covers the largest area (Fig. 6 and Fig. 9). Referring to Table 2, the wind speed in the TC is maximal on DOY 257 with 250 km/h. However, the highest VTEC disturbance took place on DOY 256 with the second-highest wind speed of 240 km. The highest VTEC deviations of different periods were observed to increase in the ionosphere over the peak intensity point of the Mangkhut typhoon. The highest VTEC variation amplitude was observed on September 13, 2018 (DOY 256), when the sustained wind speed in the TC had a second-highest value. And the lowest sea-level pressure covered the maximum area. The highest VTEC variation amplitude could happen before the max wind speed that may base on the few difference between the highest and second-highest wind speed about 10 km/h. In addition, the wind speed of 240 km/h continued for about 2 days. After that, the storm arrived at the max wind speed of 250 km/h for a short time of about a few hours. Then, the storm hit Luzon island in the Philippines, and, consequently, the earth's friction causes the wind to be decreased. This decrease in wind speed over land is obvious in Figure 1.

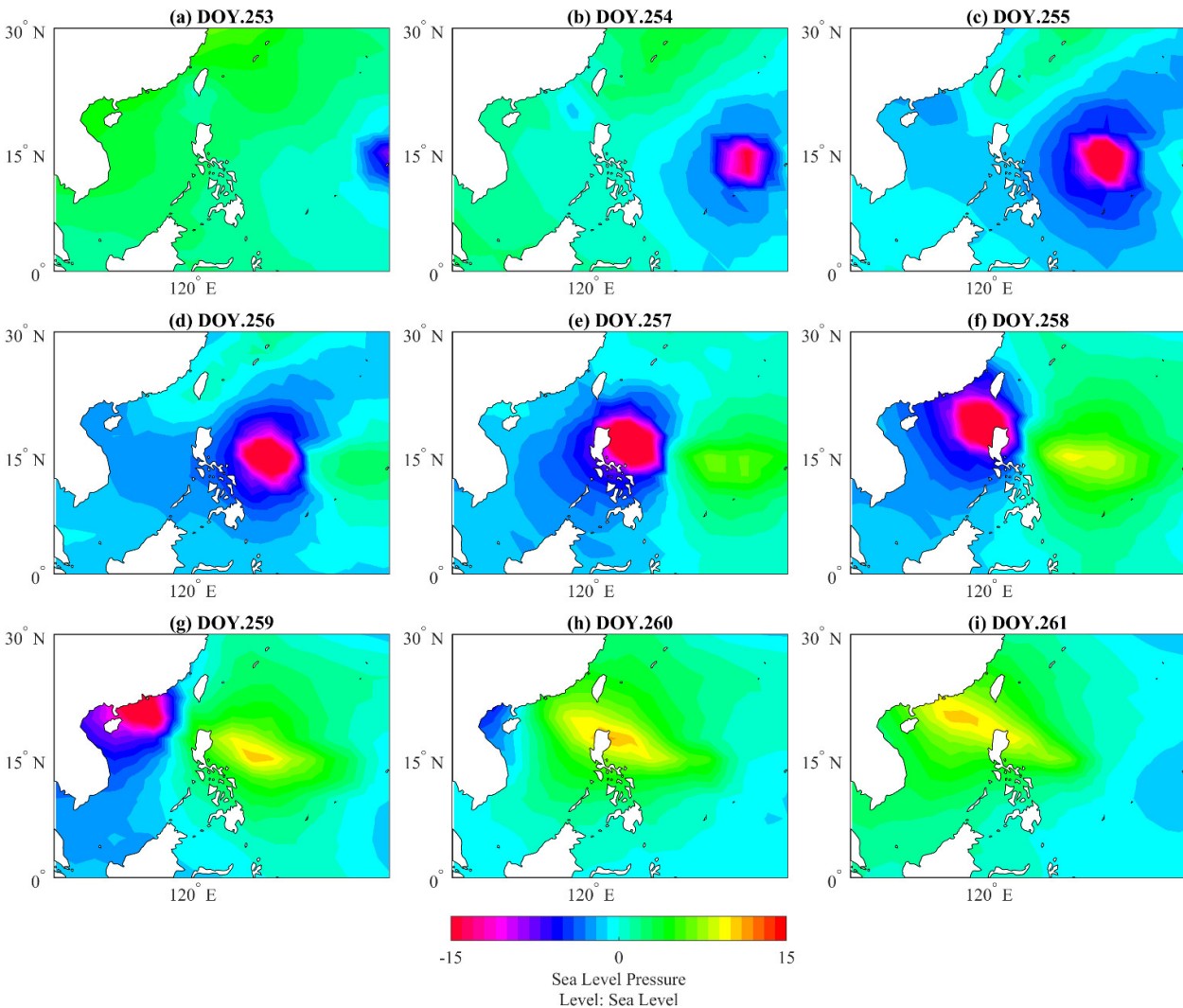

**Figure 9.** The anomaly maps of sea-level pressure on September 10–18, 2018; the panels (**a**)–(**i**) corresponds to DOY 253–DOY 261, respectively.

On the other side, the infrared satellite snapshots of typhoon cloud at selected time instances (Figure 10) shows that the location of the eye storm was close to the point of maximum speed point. But the previous day the maximum wind speed point was close to rainbands (typhoon edge), which have a high effect than the storm eye. The typhoon is a perfect example of large-scale convective cells, and the electric fields are generated by thermodynamic convection cells. In this sense, the effects on the ionosphere are stronger, where a stronger input occurs by electric currents that come from below and can affect the ionospheric electron-concentration. It is worth indicating that the typhoon's eye is cloudless, while the typhoon rainbands display more intense rain that can be associated with air-earth currents that are stronger in the rainband areas. Therefore, in the case of the ionized medium, this fact also justifies the Svensmark effect, which implies that a relevant percent of precipitation is correlated with cosmic ray flux that influences the Earth's climate by means of the production of extra condensation nuclei [61,62].

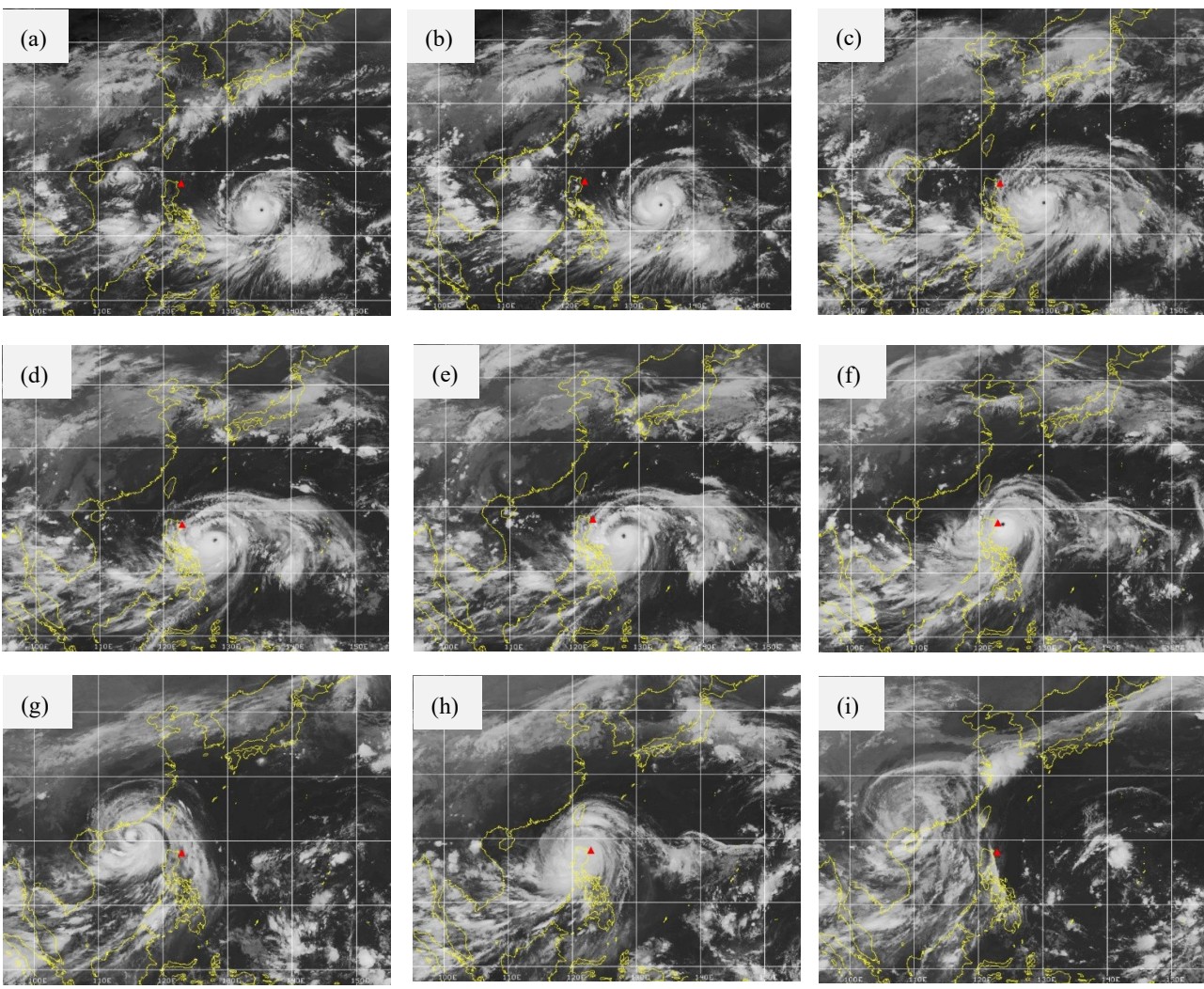

**Figure 10.** Infrared satellite snapshots of typhoon cloud for nine selected time instances (each square stand for a 100x100 area, the red triangle denotes maximum spots), (**a**) 04:30UT/DOY255, (**b**) 07:30UT/DOY255, (**c**) 07:30UT/DOY256, (**d**) 16:30UT/DOY256, (**e**) 19:30UT/DOY256, (**f**) 13:30UT/DOY257, (**g**) 22:30UT/DOY257, (**h**) 22:30UT/DOY258, (**i**) 22:30UT/DOY259.

Whereas the studies of Polyakova and Perevalova (2013) and Li et al. (2017) studies showed that during the day of the maximum wind speed value over the TC, the highest amplitude of VTEC variation was observed. In contrast, the current study showing that the highest VTEC variation amplitude was observed when the wind speed in the TC had a second-highest value. In order to interpret the possible reasons, the infrared satellite snapshots of typhoon cloud for nine selected time instances are depicted in Fig. 10. The red triangle shows the maximum spots (because of the small scale and the two maximum spots distant with about only 100 km, the red triangle could represent the max wind speed area). Panels a–i show the position and tracking of the typhoon direction from southeast to northwest and ended on September 16 (panel i). Panels c–e show different situations of September 13, when the maximum spot is far away from the eye of typhoon (it shows a relatively calm, generally cloudless area). Meanwhile, the maximum spot is close to the edge of the typhoon. In contrast, in panel f, on September 14 (the day of maximum wind speed), the maximum spot is very close to the eye of typhoon. In this sense, this fact can explain why the highest TEC variations occurred on September 13, categorized as the day

of second-highest wind speed. Subsequently, the distance away/close of the typhoon is a crucial factor for the control of the magnitude of ionospheric disturbances [38]. Our results confirm the findings by Li et al. (2017) and Chen et al. (2020) where ionospheric perturbations in the eye of the storm are fewer than those at the edges of cyclones [63]. It is worth indicating that the highest VTEC variations have happened with few hours before max wind speed that may depend on the few difference between the highest and second-highest wind speed of about 10 km/h, and the second-highest wind speed of 240 km/h was continued for about 2 days. Meanwhile, the maximum wind speed was ended within few hours after the storm arrived at the max wind speed of 250 km/h, the storm hit Luzon island, Philippines, and the wind speed going speedy decrease based on friction with the surface of the earth (Fig. 1).

Previous studies verified that the two physical mechanisms for the ionosphere could respond to powerful typhoons through gravity waves and electric fields [64]. On the same side, a transient electric field was observed through the sounding rocket and DE-2 satellite above Hurricane Debbie, but there is about 10–20 ms the electric field lasted [65]. Thus, large VTEC deviations values for several hours' duration, which are observed during powerful tropical cyclones in the northwest Pacific Ocean, are possibly be the effect of GEC produced from the thunderstorms [33]. The ionosphere regions E and F could be triggered through one of both mechanisms: the cyclone produces a low-pressure vortex system and generates gravity waves by convection cells [33]. Besides, gravity waves with vertical wavelengths could arrive at 125–200 km altitudes before going to dissipate [66]. In this regard, the gravity waves could be dissipated in the thermosphere where it could cool/heats the enclosed fluid and accelerates it in the propagation direction, and the gravity wave could transfer the energy and momentum to the thermosphere [67,68]. We suggest that the meridional winds and their resulting cold and heat waves, which depend on the poleward or equator-ward direction of the flow, with low-pressure vortex systems may contribute as a mechanism to impact the ionosphere over the powerful cyclone.

## 4. Conclusions

In this study, the ionospheric disturbances observed during powerful typhoon Mangkhut (1822) in the northwest Pacific Ocean were analyzed using the derived RIMs which were modeled from HKCORS and IGS stations data for the period before, during, and after the typhoon and its max wind speed points. The research results indicate that VTEC's highest variations observed during the second-highest wind speed with few hours before max wind speed, and the low-pressure cell covered the maximum area. These variations may be a result of the production of the GEC due to the thunderstorm and electrified convective cloud activities at the rainbands of the typhoon. The vertical conduction current produced from the GEC flows upward from the thunderstorm cloud to the ionosphere where it endorses severe variations in the electron concentrations. The variation of space weather indices indicated that the ionospheric conditions were not contributed by solar and geomagnetic activities. Besides, our research findings confirm the previous results by Li et al. (2017) and Chen et al. (2020) in that the ionospheric perturbations in the eye of the storm are fewer than those at the edges of cyclones. In addition, the critical change of the meridional wind happened on the same day of maximum ionospheric variations DOY 256. This may be a possible reason that the meridional winds and their resulting waves may contribute in one way or another to upper atmospheric coupling. This study expands evidence for more ionospheric manifestations associated with a powerful typhoon.

**Author Contributions:** Conceptualization, M.F.; Data curation, M.F., M.R., A.T., and X.R.; Formal analysis, E.Ş. and M.F.; Funding acquisition, X.Z.; Investigation, M.F. and E.Ş.; Methodology, M.F.; Project administration, X.Z. and X.R.; Resources, M.F. and M.R.; Software, M.F., E.Ş., M.A.A., and X.R.; Supervision, X.Z.; Validation, M.F., X.R., and M.R.; Visualization, M.F., E.Ş., M.A.A., B.G.M., and M.R.; Writing – original draft, M.F.; Writing – review & editing, M.F., X.Z., E.Ş., M.A.A, B.G.M A.T., X.R., and M.R. All authors have read and agreed to the published version of the manuscript.

**Funding:** This research was funded by the National Science Fund for Distinguished Young Scholars (no. 41825009), the National Natural Science Foundation of China (no. 41904026), and the Hubei Provincial Natural Science Foundation of China (no. 2020CFB824).

**Institutional Review Board Statement:** Not applicable.

**Informed Consent Statement:** Not applicable.

**Data Availability Statement:** The data presented in this study are available on request from the corresponding author.

**Acknowledgments:** The authors are very grateful to HKCORS and IGS for providing GNSS data, NASA for providing space weather indices, NOAA-PSL for providing atmospheric parameters, cooperative Institute for Meteorological Satellite Studies/the University of Wisconsin-Madison for providing infrared satellite snapshots of typhoon cloud. The authors would like to express their gratitude to Jun Chen, and Nahed Osama for reviewing this manuscript. The authors also would like to thank the Supercomputing Center of Wuhan University for permitting of use the supercomputing system in numerical calculations. The authors thank for the valuable comments of the reviewers and their promising suggestions to improve this paper.

**Conflicts of Interest:** The authors declare no conflicts of interest.

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
