# Peer review of "Analysis of Atmospheric and Ionospheric Variations Due to Impacts of Super Typhoon Mangkhut (1822) in the Northwest Pacific Ocean"

_remotesensing, doi:10.3390/rs13040661_

Round 1

Reviewer 1 Report

The paper is well designed and provides essential number of experimental data but in their conclusions it seems that the authors biased considering as the main reason for generation of the TEC anomalies AGW produced by winds. This conclusion is not supported by the TEC aptial distribution. We do not see any wave structure we see isolated spot and this implies the conslusion that most likely we are dealing with electromagnetic coupling. The recent papers on the Global Electric Circuit mechanism claim that the main generator of electricity in GEC is not the thunderstorm activity but the large convective cells, and typhoon is a prefect example of such a large scale cell. Its action leads to the local changes of the ionospheric potential what reflects on the TEC values. For the reference I recommend to authors the paper:

Ilin, N. V., Slyunyaev, N. N., & Mareev, E. A. (2020). Toward a realistic representation of global electric circuit generators in models of atmospheric dynamics. Journal of Geophysical Research: Atmospheres, 125, e2019JD032130. https://doi.org/10.1029/2019JD032130

So I consider if not to consider this effect as main source of observed TEC anomaly, at least to mention such possibility

Author Response

 Reviewer #1Comments and Suggestions for Authors:

The paper is well designed and provides essential number of experimental data…

Response: Thank you very much for your valuable comments and your interest in this paper, and we appreciate your precious time in reviewing it!

… but in their conclusions it seems that the authors biased considering as the main reason for generation of the TEC anomalies AGW produced by winds. This conclusion is not supported by the TEC spatial distribution. We do not see any wave structure we see isolated spot and this implies the conclusion that most likely we are dealing with electromagnetic coupling. The recent papers on the Global Electric Circuit mechanism claim that the main generator of electricity in GEC is not the thunderstorm activity but the large convective cells, and typhoon is a perfect example of such a large scale cell. Its action leads to the local changes of the ionospheric potential what reflects on the TEC values. For the reference I recommend to authors the paper:

Ilin, N. V., Slyunyaev, N. N., & Mareev, E. A. (2020). Toward a realistic representation of global electric circuit generators in models of atmospheric dynamics. Journal of Geophysical Research: Atmospheres, 125, e2019JD032130. https://doi.org/10.1029/2019JD032130

So I consider if not to consider this effect as main source of observed TEC anomaly, at least to mention such possibility.

Response: Thank you for your comment. We read the article you recommended and mentioned it in the first paragraph of the introduction to our work. We added the following part to Page 2, Line 53-56.“Also, a recent paper claimed that large convective cells (typhoon is a perfect example of such a large scale cell) which are the main generator of electricity in global electric circuit leads to the local changes of the ionospheric potential (IP) [12]. Both convective activity and the area covered by electrified clouds are dominant phenomena for IP parameterization. “   

We also added the following part between Line 453 and 460.“The typhoon is a perfect example of large scale convective cells, the electric fields are generated by thermodynamic convection cells. In this sense, the effects on the ionosphere are stronger where a stronger input by electric currents that come from below can affect the ionospheric electron-concentration. It is worth indicating that the typhoon’s eye is cloudless, meanwhile, the typhoon rainbands have more intense rain that may be based on air-earth currents are more intense in the rainbands areas. Therefore, in the case of the ionized medium, thus justifying also the Svensmark effect, which displays that a relevant percent of the precipitation is correlated with cosmic rays flux that influences the Earth's climate and intensifies the ionization of the connecting medium.” 

In addition, we have considered this part in the conclusion part between Line 517 and 520 as follow:“These variations may be a result of the production of the GEC due to the thunderstorm and electrified convective cloud activities at the rainbands of the typhoon. The vertical conduction current produced from the GEC flows upward from the thunderstorm cloud to the ionosphere where it endorses severe variations in the electron concentrations.” 

References

[12] Ilin, N. V., Slyunyaev, N. N., & Mareev, E. A. (2020). Toward a realistic representation of global electric circuit generators in models of atmospheric dynamics. Journal of Geophysical Research: Atmospheres, 125. https://doi.org/10.1029/2019JD032130 

*Quinn, John M., 2012. Mapping the global lithosphere: of mega-diameter meteorite impact sites within the global lithosphere, 154 pp. Solar-Terrestrial Environmental Research Institute (STERI), Lakewood, CO. 

*Quinn, John M., 2014. Global remote sensing of Earth’s magnetized lithosphere, 253 pp. Solar-Terrestrial Environmental Research Institute (STERI), Lakewood, CO. 

Reviewer 2 Report

The authors presented analysis of atmospheric and ionospheric variations during Super Typhoon Mangkhut. My recommendation is minor revision.

Line 59: The authors should add comment about hurricanes. Is clasification the same for typhoons and hurricanes?

Table 1: Typhoon should be added in row starting with 23:00 14

Line 150, Eq. (1): Should TEC be STEC?

The authors should replace in text TEC with STEC or VTEC.

line 151 (( should be (

line 152, 157: line should start from the beginning

Line 152: The authors should describe disadvatage of SLMs. Namely, signal deflection depend on space distribution of the electron density and relationship between STEC and VTEC is not so simple and used equation gives some error. If local disturbances are analized, determination of VTEC from STEC is not simple even in the case that electron density temporal and space distributions are known (that is not case in this study).

In the text and equations VTEC and STEC should be VTEC and STEC

Line 166: Approximation of single-shall (or single layer) models can provide errors because they "see" disturbances only at the considered small altitude domain. For this reason disturbances localized at other altitudes can provide errors (see for example [1] which describe errors induced by a solar X-ray flare influence which is dominant in the ionospheric D-region (below 90 km)). The authors should explain this limitation. This is important because variations of the VLF signal amplitudes (this signal are used for the lower ionospheric monitoring) during tropical cyclones (including periods around tropical depression beginnings) in several papers.

Line 228:The authors should check if a solar X-ray flare is detected. See for example https://hesperia.gsfc.nasa.gov/goes/goes_event_listings/

References: check doi numbers (in some places is doi:doi:, in some places is added link); ref. 0: Vol. should be removed; in a few references pp. should be removed…

Aleksandra Nina,

Institute of Physics Belgrade,

Univesity of Belgrade, Serbia

[1] GNSS and SAR Signal Delay in Perturbed Ionospheric D-Region During Solar X-Ray Flares, Aleksandra Nina, Giovanni Nico, Oleg Odalović, Vladimir M. Čadež, Miljana Todorović Drakul, Milan Radovanović, and Luka Č. Popović, IEEE Geoscience and Remote Sensing Letters, vol. 17, no. 7, July 2020

Author Response

Thank you very much for your valuable comments and your interest in this paper, and we appreciate your precious time in reviewing it!

Please check the attached document.

Reviewer 3 Report

Comments on the paper

Analysis of atmospheric and ionospheric variations due to impacts of Super Typhoon Mangkhut (1822) in the northwest Pacific Ocean

By Mohamed Freeshah et al.

The paper is valuable and has certainly to be published.

However, even upon considering the wide multidisciplinary perspective, it deserves a few formal improvements, aimed to strengthen the impact on a wider audience.

I refer either to formal improvements, or to the presentation style.

Let me start by a lesser misprint: at Line 151 look at a double ((

Concerning the English style, the paper is readable, although I would recommend some slight improvement, possibly made by somebody of mother English tongue.

For instance, at Line 241 concerning the term “contributed”, maybe is better “affected”.

According to my reading I had difficulty to understand a few statements, as follows:

Lines 321-322  I cannot understand the statement

Then, an indication of a critical change state in the meridional wind where the data anomalies behavior is suddenly increased from DOY 255 to DOY 256.”

What is the nominative case, what is the verb, etc.?

Lines 342-343 I cannot understand the statement

Then the storm hit Luzon island, Philippines and the wind speed going speedy decrease that based on friction with the surface of the earth.”

Lines 358-363. I find an excessively long statement. Maybe, the statement can be read and understood with some effort. But, I suggest to the authors to split the statement in different shorter statements.

Now, I enter into a few technical issues.

At Line 164 the “regularization function” is mentioned. Probably it is the term that as a standard is known as ‘’normalization constant’’ of the associated Legendre polynomial. But, how is it defined? Several conventional choices exist in the literature dealing with different applications in Earth sciences. Please specify.

The next concern is about acronyms that are extensively used and make the reading somewhat uncomfortable.

I suggest – if possible – to put a Table at the very beginning of the paper, with all acronyms in alphabetical order. The reader can scan the Table and fix out in his memory all acronyms. I find this very useful when I read a paper.

But, some acronyms are not defined.

For instance, at Line 86 what is GIM? I guess Global Ionospheric Map, but it should be stated somewhere.

What is IGS?

CORS is defined comparably later in the paper. But, the definition ought to be specified the first time the acronym is used.

How is TECU defined? All units ought always to be clearly specified, mainly concerning items that are specific of a very specific discipline.

Etc.

The introduction seems to highlight a wide perspective of the previous literature.

However, it is uncomfortable to read, as it reveals an excessive effort to compress the text in the least number of lines.

This was reasonable when journals where published only on paper. Now, these requirements are less compulsory. In any case, I guess that no serious inconvenience is implied if the introduction takes half a page or one page. The clarity of the paper would be greatly improved if some carriage returns are added when changing the subject.

The explanation of CCL and of equation (1) cannot be understood. The reader ought to go and read the related references. According to my feeling, a few better specifications ought to explain to the reader the physical assumptions, and the basic criterion that inspires the related data analysis. I wonder the authors can spend a few statement to explain this.

In general, the explanation is reasonably clear of all algorithms and methods of analysis.

Now, I enter into some physical discussion.

At Lines 204-206 the authors state: “The average of ten days before the beginning of the super typhoon on September 11, 2018, and also to exclude DOY 254 and DOY 255 from the TEC average value to avoid the possible variations in TEC as a result of the increase of geomagnetic Kp index in these two days.“ Some grammar improvement is needed. But, I enter in a physical item.

In general, as it often happens, the concern is about whether the chicken or the egg was born first.

I mean that it is generally believed that solar radiation (both electromagnetic and corpuscular) is the first driver that controls the state of the ionosphere.

On the other hand, I got convinced that a relevant and presently unrecognized observational evidence of several different kinds envisages the existence of permanent intense air-earth currents. Note that, in this way, the reasonable former working hypothesis by Gauss is contended that the mean effect of air-earth currents can be neglected, thus permitting the classical separation of geomagnetic field into external and internal origin components. This Gauss working hypothesis was never proven.

That is, I wonder that the solar activity can affect the deep Earth processes, thus producing a relevant impact on air-earth currents, that finally affect the ionosphere.

That is, when one has a large Kp index, are we sure that this effect is of mere external origin? And can we exclude that the effect has to be rather explained in terms of a more intense air-earth current that affects the ionosphere by a disturbance originated from the Earth’s interior?

I wonder an analysis can take into account such a possible dichotomy.

Note that this is NOT a criticism. Rather this is a physical discussion of the results (and methods of analysis) that are illustrated in this paper.

In this same respect, I am impressed by the repeated claim in the paper that the maximum effect is associated NOT to the eye of the typhoon, rather to the area of larger rainfall.

This is perfectly consistent with my expectation. That is, I feel convinced that the location of intense rain is to be expected to be the site of comparatively more intense air-earth currents that affect the ionosphere.

In this respect, I know that in the literature nobody explains how rain precipitation occurs, how water can coalesce around condensation nuclei, etc. I cannot enter into a very long discussion that I am going to present at a suitable time (at present, this is in preparation, and I cannot specify details by a few statements alone). Let me stress that I do expect that electric fields are crucial in determining also rain precipitation, and electric fields are generated by thermodynamic convection cells, when the medium is ionized (thus justifying also the Svensmark effect, that shows that a relevant percent of precipitation is correlated with cosmic ray flux that increases the ionization of the connecting medium).

Therefore, it is quite reasonable that, where rain is more intense, air-earth currents are more intense, hence the effects on the ionosphere is stronger, as a stronger input by electric currents that come from below can affect the electron concentration in the ionosphere.

Sometimes, in this paper, it is mentioned the role of “electric fields”. But, where the electric fields are originated? Note that this is not a criticism. In fact, I am well aware of the general philosophy of this approach.

I mean that that the authors of this paper seem to be strongly biased by looking at gravity waves - or any other kind of wave - generated by meteorological phenomena that ought to produce severe perturbations on the ionosphere.

OK, this can be one effect. Although, I wonder this is only a secondary and only occasional effect that co-exists with another steady and less imaginative phenomenon, related to a change of the always existing air-earth currents.

I believe that the evidence of steady air-earth currents is going to be, in the next future, the most revolutionary discovery of Earth science since the time of Gauss.

I do not ask the authors to revise their interpretation. I warn them from sponsoring only the fashionable viewpoint, in terms of external solar control of the ionosphere, and of mechanical waves originated from meteorology. I mean that the authors ought to consider the possibility of an effect directly on the deep Earth. The effect influences the air-earth currents that finally produce an effect on the ionosphere.

I believe that, by a few years, this alternative viewpoint is going to get a wider acknowledgement.

In this respect, I remind about a long report by the late John M. Quinn, who died on March 31, 2020. He carried out an original analysis of 6 months of magnetic record by the satellite CHAMP. He found curious “double-eye” features that can be well explained by intense air-earth currents, looking like thin “blades” of currents, flowing mainly in regions of heavy crustal fracturing. He did not agree with my interpretation, and he speculated about mysterious large magma reservoirs – that, on the other hand, ought to be very frequent on the Earth, and displaying no other geological or geophysical evidence. We discussed a lot. I do believe that his finding is the most important discovery since the time of Gauss.

Note that this same line of research and argument explains the success that is being achieved by several groups who investigated the ionospheric effects associated to earthquakes. Refer to several papers in the literature related to precursors, coseismic effects, and aftershock.

Summarizing, this paper is certainly much valuable.

I recommend to improve the presentation as per the suggestion specified above.

Then, I recommend the paper for publication, due to its relevant observational content and analysis.

References:

*Quinn, John M., 2012. Mapping the global lithosphere: of mega-diameter meteorite impact sites within the global lithosphere, 154 pp. Solar-Terrestrial Environmental Research Institute (STERI), Lakewood, CO.

*Quinn, John M., 2014. Global remote sensing of Earth’s magnetized lithosphere, 253 pp. Solar-Terrestrial Environmental Research Institute (STERI), Lakewood, CO.

NB – I have no concern about being anonymous.

Author Response

Thank you very much for your valuable comments and your interest in this paper. Also, we appreciate your precious time in reviewing! Actually, we interested in your valuable comments and gained promising ideas for our further studies, Thanks again.

Please, check the attached document.

Round 2

Reviewer 2 Report

The authors answered my questions. They should add only one sentence, remove Table 3 or add one colon if they want to keep it, and add one or two doi numbers (if they exist) and check reference 57. After these changes my recommendation is to accept the manuscript.

REMARKS:

1. Two characteristics are most important for the influence of solar X-ray flares on the ionosphere. 1. flares should occur when the observed area is sunny (daytime period), and 2. the intensity of flare. X-ray fleres are grouped into classes (A, B, C, M and X, see https://spaceweather.com/glossary/flareclasses.html ) and an assessment of their influence on the contribution of the D-region to VTEC is given in Nina et al. 2020. With intense flares, the electron density also increases above the D-region.

The authors should check which of the indicated flares occurred during the daytime in the observed area and state that they are very weak (A and/or B classes) and practically do not affect VTEC.

Because influences of these flares are not important the authors can remove Table 3. If the authors want to keep this table then should to insert another column for the flare class and keep only the flares that occurred during the day.

2. The authors should add doi numbers in references 42 (if exist) and 43, and check if reference [57] in line 351 corresponds to reference [57] in Reference.

Author Response

Thank you very much for your valuable comments. We have revised the manuscript very carefully and seriously by taking into consideration ALL of the comments and suggestions. Please, find the attachments.

Reviewer 3 Report

I forward separately my comments, as the automatic attachment which is indicated here below does not work. Sorry.

Author Response

Thank you very much for your valuable comments. We have revised the manuscript very carefully and seriously by taking into consideration ALL of the comments and suggestions. Please, find the attachment.
